# Characterization of the *WRKY* Gene Family Related to Anthocyanin Biosynthesis and the Regulation Mechanism under Drought Stress and Methyl Jasmonate Treatment in *Lycoris radiata*

**DOI:** 10.3390/ijms24032423

**Published:** 2023-01-26

**Authors:** Ning Wang, Guowei Song, Fengjiao Zhang, Xiaochun Shu, Guanghao Cheng, Weibing Zhuang, Tao Wang, Yuhang Li, Zhong Wang

**Affiliations:** 1Institute of Botany, Jiangsu Province and Chinese Academy of Sciences (Nanjing Botanical Garden Mem. Sun Yat-Sen), Nanjing 210014, China; 2Jiangsu Key Laboratory for the Research and Utilization of Plant Resources, Jiangsu Provincial Platform for Conservation and Utilization of Agricultural Germplasm, Nanjing 210014, China

**Keywords:** *Lycoris radiata*, WRKY transcription factors, expression patterns, anthocyanin biosynthesis, drought stress, methyl jasmonate treatment

## Abstract

*Lycoris radiata*, belonging to the Amaryllidaceae family, is a well-known Chinese traditional medicinal plant and susceptible to many stresses. WRKY proteins are one of the largest families of transcription factors (TFs) in plants and play significant functions in regulating physiological metabolisms and abiotic stress responses. The WRKY TF family has been identified and investigated in many medicinal plants, but its members and functions are not identified in *L. radiata*. In this study, a total of 31 *L. radiata WRKY* (*LrWRKY*) genes were identified based on the transcriptome-sequencing data. Next, the LrWRKYs were divided into three major clades (Group I–III) based on the WRKY domains. A motif analysis showed the members within same group shared a similar motif component, indicating a conservational function. Furthermore, subcellular localization analysis exhibited that most LrWRKYs were localized in the nucleus. The expression pattern of the *LrWRKY* genes differed across tissues and might be important for *Lycoris* growth and flower development. There were large differences among the *LrWRKYs* based on the transcriptional levels under drought stress and MeJA treatments. Moreover, a total of 18 anthocyanin components were characterized using an ultra-performance liquid chromatography-electrospray ionization tandem mass spectrometry (UPLC-ESI-MS/MS) analysis and pelargonidin-3-O-glucoside-5-O-arabinoside as well as cyanidin-3-O-sambubioside were identified as the major anthocyanin aglycones responsible for the coloration of the red petals in *L. radiata*. We further established a gene-to-metabolite correlation network and identified *LrWRKY3* and *LrWRKY27* significant association with the accumulation of pelargonidin-3-O-glucoside-5-O-arabinoside in the *Lycoris* red petals. These results provide an important theoretical basis for further exploring the molecular basis and regulatory mechanism of WRKY TFs in anthocyanin biosynthesis and in response to drought stress and MeJA treatment.

## 1. Introduction

Transcription factors (TFs) activate diverse signal transduction pathways and modulate the transcriptional mode of targeted genes, which improves plants to adapt to diverse environmental responses [1,2,3,4]. Among various TF families, WRKY TFs are one of the largest TF families in plants and play crucial roles in various biological processes, such as physiological metabolism, anthocyanin accumulation, and abiotic stress response [4,5,6]. WRKY proteins contain a highly conserved WRKY domain (WRKYGQK) at the N-terminus and C2H2 (C-X4–5-CX22–23-H-X-H) or C2HC (C-X7-C-X23-H-X-C) zinc finger motif at the C-terminus [5]. WRKY proteins can be classified into four main groups (Group I, II, III, and IV) based on the number of WRKY domains and the structure of zinc finger motifs. The Group I members contain two WRKY domains and a C2H2-type zinc finger. The Group II and Group III members have one WRKY domain; Group II members have a C2H2-type zinc finger and the Group III members consist of a C2HC type. The Group IV members contain an incomplete or partial WRKY domain and lack a zinc-finger motif, indicating that this group may have lost their function as WRKY TFs [4,7]. In addition, Group I can be divided into two subgroups, Ia and Ib, based on two WRKY domains at the N and C terminus [7,8]. The Group II WRKY can be further categorized into five subgroups (IIa, IIb, IIc, IId, and IIe) based on the sequence of the DNA-binding domain [8]. Group III is classified into two subgroups (IIIa and IIIb) based on the zinc-finger motif structure [9,10].

The *WRKY* gene family is widespread in many plants. For example, there are 74 *WRKY* genes in *Arabidopsis thaliana* [11], 153 *WRKY* genes in *Brassica napus* L. [12], 88 *WRKY* genes in *Phaseolus vulgaris* L. [13], 54 *WRKY* genes in *Ananas comosus* L. [14], 94 *WRKY* genes in *Sorghum bicolor* L. [15], 44 *WRKY* genes in *Liriodendron chinense* [16], 138 *WRKY* genes in *Chrysanthemum lavandulifolium* [17], and 174 *WRKY* genes in *Arachis hypogaea* L. [18]. Drought is an important abiotic stress factor that prevents plant growth and development and is a regular cause of damage. Many studies have shown that the WRKY TFs play crucial roles in the signaling and regulation of gene expression during drought stress responses [7,19,20,21]. In a recent study, *GhWRKY21* was found to negatively regulate drought stress responses in cotton [22]. *TaWRKY2*-overexpressing transgenic wheat has been observed to be improved in drought resistance [23]. In *Arabidopsis*, *AtWRKY11*, *AtWRKY17*, *AtWRKY28*, *AtWRKY30*, and *AtWRKY63* have been shown to play positive regulation roles in drought stress responses, while *AtWRKY46*, *AtWRKY54*, and *AtWRKY70* participated in plant growth and play a negative role in drought tolerance regulation [24,25,26,27,28]. Moreover, the overexpression of *GhWRKY1*, *IgWRKY50*, *IgWRKY32*, *MfWRKY41*, *TaWRKY75*, *OsWRKY114*, *ZmWRKY106*, and *TaWRKY133* enhances drought tolerance in transgenic *Arabidopsis* plants [29,30,31,32,33,34,35,36].

The *WRKY* genes could be up-regulated under methyl jasmonate (MeJA) treatment, suggesting that they are involved in response to abiotic stress, as well as in jasmonic acid (JA) signaling pathways in plants [37]. In Populus, most *WRKY* genes were induced by MeJA and SA treatments [38]. The expression level of *ScWRKY3* in sugarcane was increased under salt, drought stresses, and abscisic acid (ABA) treatment but inhibited by salicylic acid (SA) treatment and MeJA treatment [39]. Differently, the expression level of *ScWRKY5* was induced by salt, drought stresses, as well as SA and ABA treatments [40]. Furthermore, overexpressing *HbWRKY82* of *Hevea brasiliensis* improved drought and salt tolerance and decreased sensitivity to ABA in *Arabidopsis* [41]. Recently, reports have shown that WRKY TFs have a visible correlation with the regulation of anthocyanin biosynthesis. For example, *MdWRKY11* participates in anthocyanin accumulation by affecting *MdMYB* and *MdHY5* in apples [42]. *PyWRKY26* targeted the *PyMYB114* promoter to regulate anthocyanin biosynthesis and transport in red-skinned pears [43]. *PbWRKY75* may promote *PbMYB10b* expression and activate anthocyanin biosynthetic genes *DFR* and *UFGT* to promote anthocyanin accumulation [44]. Moreover, *StWRKY13* could significantly activate anthocyanin biosynthetic genes *StCHS*, *StF3H*, *StDFR*, and *StANS* to promote anthocyanin biosynthesis in potato tubers [45]. In a previous study, *McWRKY71* interacted with *McMYB12* and directly activated *McANR* to participate in the regulation of proanthocyanidin biosynthesis [46]. In apples, *MdWRKY75* mainly activate the promoter of *MdMYB1* and induced the accumulation of anthocyanin [47].

*L. radiata* is a perennial herbaceous flower with good ornamental characteristics and barren-resistant, water-saving, and drought-resistant properties. Plants in the genus *Lycoris* have been utilized in traditional medicine preparation and more than 110 potent structurally distinct Amaryllidaceae alkaloids were isolated or identified for extensive pharmacological and phytochemical investigations [48]. For example, according to the Compendium of Materia Medica, *Lycoris* plants are described as potent antidotes to poison, effective agents to alleviate pain and relieve inflammation, and diuretic drugs [49]. Amaryllidaceae alkaloids from *L. radiata* bulbs were traditionally used for treating sore carbuncle, neurodegenerative diseases, poliomyelitis, suppurative wounds, and ulcers [50,51,52]. In addition, *L. radiata* is an important greening material for energy-saving medicinal construction and a genetic treasure trove for the drought resistant breeding of *Lycoris* and other medicinal plants. Thus, it is of great value for deeply studying the drought resistance of *Lycoris*. At present, amaryllidaceae alkaloid and anthocyanin biosynthesis as well as sucrose degradation have been identified in *Lycoris* plants through transcriptome sequencing [53,54,55,56,57,58]. In this work, 31 *LrWRKY* genes were identified based on our previous *L. radiata* transcriptome data [53] and their motif pattern and phylogenetic relationship between *Arabidopsis* and *L. radiata* were analyzed. Subcellular localization analysis revealed that these LrWRKY proteins were mainly localized in the nucleus. The transcriptome datasets and the qRT-PCR results showed that the expression levels of *LrWRKY* genes changed in different tissues as well as under MeJA treatment and drought stress. To better comprehend the molecular mechanism of petal color formation, we identified *LrWRKYs*, the key structural genes and metabolites that are involved in anthocyanin biosynthesis, combining gene expression and anthocyanin metabolome analyses. Moreover, the probable protein–protein interaction (PPI) of LrWRKYs were also predicted. Our results provide important insights into the *LrWRKY* genes in *L. radiata* and lay a foundation for the further investigation of *LrWRKY* genes functioning in the biological pathway.

## 2. Results

### 2.1. Identification and Characterization of LrWRKY Proteins in L. radiata

TWe used HMMER 3.0 and BLASTP to predict putative LrWRKY protein sequences in the *L. radiata* transcriptome database with an E-value threshold of <1 × 10^−5^. All the candidate sequences were confirmed in NCBI and with SMART to further identify the conserved complete WRKY domains. Thirty-one LrWRKY proteins ranged from LrWRKY1 to LrWRKY31 were identified and their basic characteristics are shown (Appendix A). The coding sequence lengths of *LrWRKY* genes were varied from 459 bp to 1830 bp. The LrWRKY proteins ranged from 153 to 610 amino acids in size, from 17.41 to 66.53 kDa in molecular weight, and from 4.80 to 9.83 in their isoelectric points. The subcellular localization of the LrWRKY proteins was predicted by using ProtComp 9.0, Plant-mPLoc, and WOLF PSORT. It showed that most of the LrWRKY proteins were probably located in the nucleus, and few LrWRKY proteins were located in the endoplasmic reticulum and chloroplast. Sixteen LrWRKY proteins showed the highest sequence similarity with *Asparagus officinalis* WRKY proteins after homologous alignments. LrWRKY4 and LrWRKY24 had the highest sequence similarity with the *Narcissus hybrid* cultivar homologous protein. LrWRKY13 had the highest sequence identity with the *Narcissus tazetta* subsp. homologous protein.

### 2.2. Phylogenetic Analysis and Classification of LrWRKY Proteins

To investigate the evolutionary relationships of the LrWRKY and *Arabidopsis* WRKY (AtWRKY) proteins, a phylogenetic tree was constructed (Figure 1). The results showed that 31 LrWRKY proteins were clustered into three subfamilies consistent with the tree topology and classification of AtWRKY proteins. Meanwhile, LrWRKY proteins could be categorized into Group I, Group IIa, Group IIb, Group IIc, Group IId, Group IIe, and Group III. Ten LrWRKY proteins were classified as Group I, containing two WRKY domains and one C2H2 zinc-finger motif. A total of 18 LrWRKY proteins included one WRKY domain and one C2H2 zinc-binding motif, which were considered to be Group II. Three *LrWRKY* genes were categorized to Group III, consisting of a single WRKY domain and a C2CH zinc-binding motif. According to the *Arabidopsis* WRKY subgroup classification, the LrWRKYs in Group II were further subdivided into five subgroups, including groups IIa (1), IIb (2) IIc (6), IId (5), and IIe (4) (Figure 1). These results revealed that WRKY proteins existed in various plant species before their divergence and then independently evolved in each species.

### 2.3. Multiple Sequence Alignment, Conserved Motifs, and WRKY Domains of LrWRKY Proteins

To further understand the diversification and conservation of LrWRKY proteins, the conservative motifs of LrWRKYs were predicted by utilizing MEME software. We found that LrWRKY proteins contained a different number of the 10 conserved motifs (Figure 2 and Appendix A). Motif 1 and Motif 2 were the two highly conserved motifs present in most LrWRKY proteins except for LrWRKY6, LrWRKY9, LrWRKY11, LrWRKY21, LrWRKY25, and LrWRKY28. The WRKY in the same group contained similar motif structures. Motif 1 and Motif 4 were WRKY domains; Motif 1 was distributed the whole groups, whereas Motif 4 was present in Group I and IIc. Motif 10 was unique to Groups IId, whereas Motif 8 to Group I and Motif 9 to Group IIe. The clustered LrWRKY pairs, such as LrWRKY1/2, LrWRKY6/28, LrWRKY12/18, and LrWRKY22/23, displayed a highly similar motif distribution (Figure 2). In addition, the same subgroup of LrWRKY proteins has similar motifs, indicating that the protein structure of this family was conserved. The conservative motif arrangement and phylogenetic analysis of the same group proteins can be used as an important basis for categorizing protein.

In addition, conserved amino acids in the WRKY domain analysis via multiple sequence alignments of the LrWRKY proteins were performed (Figure 3). Among the 31 LrWRKY proteins, 9.55% of the proteins have homology in the “WRKYGQK” core domain, which is used for defining classification of the WRKY TFs, while only two proteins (LrWRKY1 and LrWRKY2) have “WRKYGKK” domain (Figure 3). The LrWRKY proteins also had the CX4–7-C-X22–23-H motif, forming C2HC or C2H2 zinc-finger structures.

### 2.4. Multiple Sequence Alignment, Conserved Motifs, and WRKY Domains of LrWRKY Proteins

To further characterize the functions of LrWRKY TFs, GO annotation and enrichment analysis were performed for all the LrWRKYs. The LrWRKY proteins were annotated to three main GO categories (biological process, cellular component, and molecular function), including 55 biological process terms, nine molecular function terms, and one cellular component terms. The top 20 GO terms of level two were visualized (Figure 4). In biological process terms, 31 LrbWRKY proteins were implicated in the regulation of transcription. Most LrWRKYs regulate a defense response to bacterium, response to chitin, defense response to fungus, metal ion binding, regulation of defense response, response to jasmonic acid and SA, as well as response to water deprivation. In cellular component terms, 31 LrWRKY proteins were components of the nucleus. In molecular function terms, 31 LrWRKY proteins were categorized as exhibiting DNA-binding transcription factor activity and sequence-specific DNA binding, respectively (Figure 4).

### 2.5. Multiple Sequence Alignment, Conserved Motifs, and WRKY Domains of LrWRKY Proteins

As the transcriptome analysis of different tissues (root, leaf, and bulb) has been revealed in *Lycoris longituba* [57], we then searched the orthologous genes of *LrWRKY* by using local blast within *L. longituba* transcriptome data; 31 orthologous genes of *LrWRKY* were found in *L. longituba* (Appendix A). As indicated in Appendix A, some *LrWRKY* genes exhibited differential expressions in the three tissues, whereas other *LrWRKY* genes showed similar expression patterns in diverse tissues. For example, two *LrWRKYs*, including *LrWRKY14* and *LrWRKY27*, were relatively highly expressed in leaves, whereas *LrWRKY22* and *LrWRKY23* were preferentially expressed in roots. *LrWRKY3*, *LrWRKY13*, *LrWRKY16*, and *LrWRKY30* had the highest relative expression levels in bulb. To further elucidate the biological function of LrWRKY proteins, qRT-PCR was utilized to determine the spatial specificity expression pattern of 31 *LrWRKY* genes in eight *L. radiata* organs. As indicated in Figure 5A, some *LrWRKY* genes exhibited differential expression in the eight tissues, whereas other *LrWRKY* genes showed similar expression patterns in diverse tissues, which could be attributed to the functional differentiation of *LrWRKY* genes during plant development. For example, three *LrWRKYs* (i.e., *LrWRKY11*, *LrWRKY12*, and *LrWRKY17*) were relatively highly expressed in leaves. *LrWRKY22* and *LrWRKY23* were preferentially expressed in petals. *LrWRKY29* showed high expression levels in gynoecium, whereas *LrWRKY30* had relatively high expression levels in bulb. In addition, five *LrWRKYs* (i.e., *LrWRKY2*, *LrWRKY5*, *LrWRKY13*, *LrWRKY16*, and *LrWRKY19*) were highly expressed in stamens tissues. In particular, *LrWRKY4*, *LrWRKY6*, *LrWRKY10*, *LrWRKY18*, *LrWRKY24*, and *LrWRKY31* were predominantly expressed in roots. *LrWRKY15* was relatively highly abundant in seeds. *LrWRKY1* and *LrWRKY25* had the highest relative expression levels in flower stalks. Among all the tissues, the fewest *LrWRKY* members were expressed the most in the roots and leaves. Conversely, some genes were not expressed specifically. We noticed that similar expression patterns for *LrWRKY7*, *LrWRKY9*, *LrWRKY10*, *LrWRKY24*, *LrWRKY28*, and *LrWRKY31* in different tissues suggest possible redundancy. These results suggested that *LrWRKYs* might play important functional roles in the growth and development of *L. radiata*.

On the basis of tissue-specific expression, the expression of *LrWRKYs* was further observed at the flowering developmental stages using previously published RNA-seq data (Figure 5B). In *L. radiata* flowering development, FB, FL1, FL2, and R indicated the initial stage of flower—bud differentiation, partially opening flower, fully opened flower, and senescent flower stage, respectively (Figure 5B). Over half of the *LrWRKY* genes were expressed lower at the FB stage and then markedly increased at the R stage. In addition, the expression of *LrWRKY15* and *LrWRKY31* exhibited a gradual increase during flower development. Remarkably, the expression of *LrWRKY16* and *LrWRKY26* genes exhibited opposite trends in the flowering developmental stages, indicating they may perform diverse functions. However, the expression of *LrWRKY8*, *LrWRKY9*, *LrWRKY19*, and *LrWRKY30* were relatively higher at the early stages of flower development.

### 2.6. Expression Patterns of LrWRKYs in Response to Drought Stress and MeJA Treatment

We subsequently examined the gene expression profiles of *LrWRKY* in the roots and leaves of one-year-old *L. radiata* seedlings that were subjected to drought stress for 24 h. As shown in Figure 6A, the responses of *LrWRKYs* were different between individuals after drought treatment. In the *L. radiata* leaves and roots, the majorities of *LrWRKY* genes were down-regulated and showed statistically significant higher expression patterns after 24 h of drought stress compared to the control (Figure 6A). Nonetheless, the expression of *LrWRKY5*, *LrWRKY22*, and *LrWRKY23* reached the highest level under drought stress in *L. radiata* roots. Moreover, the expression levels of *LrWRKY2*, *LrWRKY6*, *LrWRKY28*, *LrWRKY30*, and *LrWRKY31* genes gradually down-regulated after drought stress in *L. radiata* roots. In the *L. radiata* leaves, *LrWRKY4* and *LrWRKY8* were significantly up-regulated under drought treatment. Besides, the expression levels of *LrWRKY1*, *LrWRKY9*, *LrWRKY16*, *LrWRKY19*, *LrWRKY20*, *LrWRKY21*, *LrWRKY25*, and *LrWRKY26* gradually down-regulated under drought treatment in leaves (Figure 6A). These results suggest that *LrWRKY* genes regulate plant physiological processes.

Previous study of *Lycoris aurea* transcriptome sequencing has revealed that MeJA treatment could induce the expression of *LaWRKY* genes [54]. Thus, the orthologous genes of *LrWRKY* in *L. aurea* transcriptome data were also searched. All the 31 *LrWRKY* genes were found to have orthologous transcripts in *L. aurea* transcriptome (Appendix A). Among them, eight homologous *LrWRKYs* were up-regulated, while nine homologous *LrWRKY* genes were suppressed with MeJA treatment for 6 h (Appendix A). In addition, homologous genes of *LrWRKY5* and *LrWRKY17*, *LrWRKY15*, and *LrWRKY16* exhibited no expression changes after MeJA treatment. Further, the expression patterns of the *LrWRKYs* tested using qRT-PCR and most *LrWRKYs* expression levels were up-regulated by MeJA treatment and varied over time (Figure 6B). The transcription level of *LrWRKYs* was up-regulated after 6 h, whereas it was down-regulated at 12 h. Specifically, the expression levels of nine genes (*LrWRKY6*, *LrWRKY11*, *LrWRKY12*, *LrWRKY13*, *LrWRKY15*, *LrWRKY18*, *LrWRKY24*, and *LrWRKY28*) increased rapidly under MeJA treatment, while those of the other ten genes (*LrWRKY5*, *LrWRKY8*, *LrWRKY16*, *LrWRKY19*, *LrWRKY20*, *LrWRKY22*, *LrWRKY23*, *LrWRKY25*, *LrWRKY26*, and *LrWRKY31*) were lower under MeJA treatment than that in the control. In addition, the expression profiles of some genes changed dramatically in specific time. For instance, the expression performance of *LrWRKY7* and *LrWRKY9* at 24 h were lower than those in the control but were significantly unregulated at 36 h. The expression trends of *LrWRKY10*, *LrWRKY11*, *LrWRKY14*, and *LrWRKY22* were consistent with the transcriptome data. There were certain divergences between the actual examined and transcriptome data, which can be demonstrated by the individual divergences between the material transcriptome data and the qRT-PCR analysis. In particular, the expression pattern of *LrWRKY6*, *LrWRKY12*, *LrWRKY18*, and *LrWRKY28* up-regulated rapidly at 6 h. After 24 h of MeJA treatment, the expression pattern of *LrWRKY17* and *LrWRKY21* up-regulated rapidly. From these data, it can be speculated that MeJA regulates the expression pattern of the *LrWRKY* genes. These findings revealed that *LrWRKYs* displayed different expression patterns in response to MeJA hormone treatment and potentially participated in developmental processes via hormone signaling pathways in *Lycoris*.

### 2.7. Identification of the Main Anthocyanin Pigments in L. radiata Petals

To detect the metabolic mechanism of the red petal color phenotype in *L. radiata*, the total anthocyanins in the petals were measured [53]. To further ascertain the metabolic mechanism of the *L. radiata* petals, the anthocyanin metabolites were detected using UHPLC-ESI-MS/MS; 18 anthocyanin compounds were identified and quantified in the *L. radiata* petals (Figure 7A). We found that the pelargonidin-3-O-glucoside-5-O-arabinoside content was the highest in the *L. radiata* petals, followed by Cyanidin-3-O-sambubioside (Figure 7A). Given that these two compounds may play a key role in red petal color formation in *L. radiata*, we propose that these anthocyanins were the main compounds responsible for the red coloration of petal margins.

### 2.8. Integrating Related LrWRKY Genes and Metabolites in Anthocyanin Biosynthesis Pathway

WRKY TFs are important regulators of anthocyanin accumulation and they characteristically act by controlling the structural gene expression in the anthocyanin biosynthesis pathway. To enunciate the relationship between *LrWRKYs* and anthocyanin biosynthesis in *L. radiata*, we established a co-expression network of *LrWRKYs* and the anthocyanin metabolites (Figure 7B,C and Appendix A). The PCC between the expression performance of *LrWRKYs* and the anthocyanin metabolites was further calculated. The results implied that eight *LrWRKYs* (*LrWRKY1*, *LrWRKY3*, *LrWRKY6*, *LrWRKY7*, *LrWRKY13*, *LrWRKY15*, *LrWRKY17*, and *LrWRKY27*) positively regulated the anthocyanin metabolites, whereas four *LrWRKYs* (*LrWRKY16*, *LrWRKY22*, *LrWRKY25*, and *LrWRKY29*) negatively regulated the anthocyanin metabolites. Furthermore, pelargonidin-3-O-glucoside-5-O-arabinoside was strongly correlated with the expression of *LrWRKYs*, indicating that pelargonidin-3-O-glucoside-5-O-arabinoside play a significant role in the red petal color in *L. radiata*. Among these *LrWRKYs*, the expression level of *LrWRKY3* and *LrWRKY27*, indicated a remarkable positive correlation with the pelargonidin-3-O-glucoside-5-O-arabinoside content in the samples (PCC > 0.9, Figure 7B,C and Appendix A), suggesting that *LrWRKY3* and *LrWRKY27* may have an essential role in the pelargonidin accumulation.

The correlation between *LrWRKYs* and differently expressed genes (DEGs) of anthocyanin biosynthesis in *L. radiata* could be divided into three main clusters (Figure 8A and Appendix A). In Cluster II, several *LrWRKY* genes (*LrWRKY1*, *LrWRKY3*, *LrWRKY5*, *LrWRKY7*, *LrWRKY13*, *LrWRKY15*, *LrWRKY17*, *LrWRKY24*, *LrWRKY26*, and *LrWRKY27*) showed strong correlations with *phenylalanine ammonia lyase* (*PAL*), *cinnamate 4-hydroxylase* (*C4H*), *chalcone synthase* (*CHS*), *dihydroflavonol reductase* (*DFR*), *flavonoid 3′-hydroxylase* (*F3*′*H*), and *leucoanthocyanidin reductase* (*LAR*) genes in the anthocyanin pathway. In contrast, the *LrWRKYs* in Cluster I showed a low correlation with anthocyanin biosynthetic pathway genes in *L. radiata* and only correlated strongly with upstream one *PAL* and one *4-coumarate:CoA ligase* (*4CL*) genes. Some *LrWRKYs* in Cluster III showed a strong negative correlation with the pathway genes. The co-expression analysis showed that the anthocyanin pathway genes were relevant to *LrWRKYs* in Cluster II and that *LrWRKY3*, *LrWRKY7*, *LrWRKY26*, and *LrWRKY27* were selected for their modulatory potential in anthocyanin biosynthesis. In particular, we observed that the identified anthocyanin biosynthesis DEGs and key transcription factors were significantly correlated with nine anthocyanins, including Pelargonidin-3-O-glucoside-5-O-arabinoside, Malvidin-3-O-(6″-O-malonyl) glucoside, Delphinidin-3,5,3′-Tri-O-glucoside, Delphinidin-3-O-glucoside, Cyanidin-3-O-(6″-O-p-Coumaroyl) glucoside, Cyanidin-3-O-(6″-O-p-coumaroyl-2‴-O-sinapoyl)sambubioside-5-O-(6″″-O-malonyl) glucoside, Cyanidin-3-O- (6″-O-feruloyl) glucoside-5-O-glucoside, Peonidin-3-O-(6″-O-p-Coumaroyl) glucoside, and Peonidin-3-O-(6″-O-caffeoyl) glucoside (Figure 8B and Appendix A). Some crucial *LrWRKYs* involved in anthocyanin biosynthesis, such as *LrWRKY3*, *LrWRKY7*, *LrWRKY21*, *LrWRKY26*, *LrWRKY27*, and *LrWRKY28* were positively related to the accumulation of nine anthocyanins and anthocyanin enzyme-encoding DEGs including *anthocyanidin synthetase* (*ANS*), *4CL*, *CHS*, *C4H*, *chalcone isomerase* (*CHI*), *DFR*, *F3′H*, *LAR*, and *PAL*. These results revealed that the *LrWRKY* genes and anthocyanin biosynthesis genes might be involved in the petal red color in *L. radiata*.

### 2.9. PPI Networks of LrWRKY Proteins

Based on the orthologs of *Arabidopsis* (Appendix A), the STRING database predicted that numerous LrWRKY proteins associate with each other (Figure 9; Appendix A), suggesting that the WRKY proteins binding activity is based on the homodimer or heterodimer formation among WRKY proteins [59]. Generally, we predicted several key interactions (Figure 9, Appendix A). We identified eight high-confidence interacting proteins in the *Arabidopsis* WRKY family: ASC6 (aminocyclopropane-1-carboxylic acid synthase 6) and GUN5 (genomes uncoupled 5) proteins involved in abiotic stress responses; MPK1 (mitogen-activated protein kinase 1), MPK3, and MPK4 proteins involved in plant responses to pathogens and stresses; and VQ motif-containing (VQ) proteins such as MKS1 (Meckel syndrome, type 1), SIB1 (sigma factor binding protein 1), and SIB2 involved in regulating plant defense responses. In addition, the LrWRKY10 is highly homologous to Arabidopsis WRKY33, indicating that it may have potentially stronger interactions with most plant defense proteins (MPK3, MPK4, MKS1, and SIB1). Similarly, the LrWRKY13, LrWRKY21, and LrWRKY27 protein is highly homologous to *Arabidopsis* WRKY40, WRKY70, and WRKY6, which are presumed to have a strong interaction with the internal members of the WRKY family. These findings partially validate the anticipated interaction networks and suggest that they may have comparable roles in *L. radiata*.

### 2.10. Subcellular Localization of LrWRKY Proteins

Based on prediction analysis, most LrWRKY proteins were predicted as localized in the nucleus, whereas few of the LrWRKYs were located in the chloroplast and/or endoplasmic reticulum (Appendix A). Subsequently, certain LrWRKY proteins fused to GFP were transiently expressed in *N. benthamiana* epidermal cells using 35S-GFP construct as a positive vector to validate their subcellular localization. The results confirmed that LrWRKY3, LrWRKY10, LrWRKY13, LrWRKY21, and LrWRKY27 proteins were localized in the nucleus (Figure 10). Thus, these fluorescent microscopy data confirm that most LrWRKYs are nucleoproteins based on their targeting and translocation to the nucleus.

## 3. Discussion

### 3.1. Characterization of the WRKY Gene Family in L. radiata

In plants, WRKY TFs are one of the largest transcription families, playing an important role in plant physiological processes, diverse biotic and abiotic stress responses, and secondary metabolite synthesis in different crop species, including *Arabidopsis*, Chinese cabbage, peanut, pepper, rice, and tomato [5,7,8,18,60,61,62,63,64]. WRKY TF families have been reported in many plants, but little is known concerning the functions of the WRKY TF in *L. radiata*. In this study, we identified 31 *LrWRKY* genes through homolog search and domain analysis utilizing previously published transcriptome data (Appendix A; [53]). The LrWRKYs had similar properties in terms of the encoding amino acids number, isoelectric point, and motifs to WRKYs in *A. thaliana* and other plants [62]. In our research, we combined the phylogeny analysis with full-length protein sequences and the classification of *Arabidopsis* WRKY groups to categorize the WRKY groups in *L. radiata* [10]. Generally, highly conserved WRKYs will group together and perform similar or identical molecular functions, pertaining to the same WRKY group (Figure 1). In general, the WRKY proteins of *L. radiata* classified into three groups (containing Group I, Group II (II-a, II-b, II-c, II-d, II-e), and Group III) contingent on the conserved types of WRKY domains and their zinc-finger domains, which were similar to other plants [60,61,62,63,64].

Previous studies proposed four main WRKY TF lineages in flowering plants, namely Group I + IIc, Group IIa + IIb, Group IId + IIe, and Group III, which accurately reflected the evolution of the WRKY TF family members (Figure 1; [65]), which was also verified for *L. radiata*. For example, the Group IIa members and Group IIb members were in the same evolutionary branch and the Group IId members and Group IIe members were grouped into the same branch, suggesting that these subgroups arose indirectly from a common ancestor. Out of the 31 members, 29 were properly conserved in the “WRKYGQK” domain. The substitution of glutamine occurs instead of lysine in two WRKY heptapeptide domains (Figure 3). This “WRKYGKK” domain sequence is reported to be a major variant in many studies [60,61,62,63,64,65,66]. The mutation in the WRKYGQK conserved domain mainly occurs from “Q” to “K” amino acid. For example, LrWRKY1 and LrWRKY2 replaced “Q” by “K” as their respective domains were also reported in *Camelina sativa* [66]. The same kind of substitution was observed in the *WRKY* gene family of *Chrysanthemum lavandulifolium* [17]. The motif of *LrWRKY* genes on the same branch commonly has a similar domain component. These results can be used as an important basis for WRKY protein classification (Figure 2). These motifs are associated with transcriptional activity and protein–protein interactions of target genes [67]. Therefore, the characterizations and functions of LrWRKY TFs can be appraised by analyzing conserved motif information. All the LrWRKYs had more than two conserved motifs, coincident with other plant species reports [68,69]. Each group member had the same or similar motif constitute, suggesting that the same group WRKY TFs have same or similar protein structures and biological functions.

### 3.2. The WRKY Gene Expression Patterns Facilitate Their Functional Analysis

We methodically analyzed the expression patterns of *LrWRKY* TFs in various tissues to investigate their physiological role in *L. radiata* biological processes. The homologous genes with similar expression profiles are conserved due to the dosage effect, whereas the homologous genes with different expression profiles are retained by functionalization and subfunctionalization [70]. We used transcriptome data from different tissues of two *Lycoris* species to determine the WRKY family gene expression (Figure 5 and Appendix A). In the present study, the *LrWRKYs* of *Lycoris* were expressed differentially in the leaves, roots, flowers, and stems. The *WRKY* TF genes play an important role in plant development. This indicated that *LrWRKYs* play a critical role in the growth and development of *Lycoris*. The *LrWRKY* genes were expressed in at least one tissue and their expression patterns varied markedly among the different tissues. As shown in Appendix A, the *LrWRKY* genes in roots were higher than those in the leaves and bulb. This study also analyzed the differences in eight tissues of *L. radiata* (Figure 5A). Notably, six *LrWRKY* genes displayed the highest expression patterns in roots. Three *LrWRKY* genes were highly expressed in the leaves, whereas one *LrWRKY* gene was highly expressed in the gynoecium, or seed, or bulb. Five *LrWRKY* genes and two *LrWRKYs* genes were highly expressed in stamens and in flower-stalk tissues, respectively. These tissue-specific expression patterns propose that *LrWRKYs* may be involved in *Lycoris* tissue-specific developmental and signal transduction processes. As reported, the *AtWRKY70* gene has significantly contributed to *Arabidopsis* root development. The experimental results proclaimed that the *wrky70* mutant had longer roots than the wild type [28,71]. Similarly, the orthologous protein of *AtWRKY70*, named *LrWRKY21*, was specifically expressed in the roots (Figure 5A). This research indicated *LrWRKY21* may be involved in regulating the *L. radiata* roots growth and development. Moreover, *AtWRKY75* also regulate the early accumulation of anthocyanin and increase the length and number of lateral roots and root hairs [72]. The *LrWRKY3* was orthologous genes of *AtWRKY75*, which has difference expression levels in roots and flower (Figure 5). These results displayed the significant contribution of *LrWRKY3* and *LrWRKY21* in the development of roots. Moreover, *LrWRKY29* is significantly expressed in the gynoecium at the *L. radiata* flowering stage and has a close homology with *Arabidopsis AtWRKY2*, suggesting that *LrWRKY29* play an important role in the flowering stage of *Lycoris*, as *Arabidopsis* homologs gene *AtWRKY2* are the key regulators of pollen development [73].

In addition to playing a momentous role in plant development, the WRKY TFs also play critical roles in plant abiotic stress responses [21]. In the present research, *AtWRKY75* gene facilitates leaf senescence and flowering in *Arabidopsis* [74,75]. A recent study showed that *PdeWRKY75*, which is orthologous with *AtWRKY75*, is involved in the manipulation of multiple biological processes by regulating hydrogen peroxide content in poplar [76]. In the current research, the expression level of *LrWRKY3* (homolog of *AtWRKY75*) was increased under drought stress, suggesting that *LrWRKY3* may participate in the drought stress response mechanism of *Lycoris* by regulating the activity of some hormone-like proteins and enhancing the *Lycoris* drought tolerance (Figure 6A). The overexpression of *AtWRKY8*, *AtWRKY28*, and *AtWRKY17* enhance drought or salt tolerance stress in *Arabidopsis* [27,77]. Similarly, the expression pattern of *LrWRKY11* (homolog of *AtWRKY8*, *AtWRKY28*, and *AtWRKY17*) can also be induced after drought treatment and interact with some regulatory factor or unknown binding protein to regulate the expression pattern of target genes, indicating that *LrWRKY11* plays an important role in drought stress tolerance in *L. radiata*. The expression level of *AtWRKY40* induces drought stress [78] and is consistent with the *LrWRKYs* expression dataset results, which showed that *LrWRKY13* (homolog of *AtWRKY40*) is involved in the drought stress response. *AtWRKY33* plays a critical role in broad plant stress responses in *Arabidopsis* [79], suggesting that drought stress induces *LrWRKY10* (homolog of *AtWRKY33*). Therefore, *LrWRKY3*, *LrWRKY10*, *LrWRKY11*, and *LrWRKY13* have played potential functions in the drought stress response. The results indicate that these genes potentially are involved in the drought resistance of *L. radiata*. Nonetheless, further experimental studies should be carried out to enunciate the accurate regulatory mechanism through which *LrWRKYs* respond to drought stress.

MeJA is an importantregulator in plant growth, stress response, and secondary metabolism in both angiosperms and gymnosperms [80,81]. Many studies found that anthocyanin biosynthesis and many MeJA-responsive TFs are regulated by MeJA signaling pathway [82,83,84,85]. For example, *AtWRKY11* (homolog of *LrWRKY6*), *AtWRKY17*, and *AtWRKY70* (homolog of *LrWRKY21*) are essential components of the confrontational action in JA and the SA-mediated signaling pathway [28,86]. In addition, *WRKY75* (homolog of *LrWRKY3*) regulated the JA-mediated signaling pathway to modulate defense responses in *A. thaliana*. Under normal conditions, WRKY75 interacted with Jasmonate ZIM domain (JAZ) protein to inhibit the JA signaling pathway [87]. *PpWRKY46* and *PpWRKY53* contribute to MeJA-primed defense by regulating the energy metabolism in peaches [88]. In tomato, *SlWRKY37* positively regulates jasmonic acid- and dark-induced leaf senescence [89]. This study used publicly available transcriptome data and qRT-PCR to study the *LrWRKY* gene expression under MeJA induction (Figure 6B and Appendix A). *LrWRKY10*, *LrWRKY11*, *LrWRKY14*, and *LrWRKY22* showed the same expression pattern and the expression levels of *LrWRKY6*, *LrWRKY12*, *LrWRKY18*, and *LrWRKY28* increased significantly at 6 h under MeJA treatment. In summary, our findings provide a valuable resource for select candidate *LrWRKY* genes, which can facilitate the further functional studies of *LrWRKYs* involved in various biological stresses.

### 3.3. LrWRKYs Are Related to L. radiata Petal Color Change

Anthocyanins are among the key metabolites of petal color formation in plants, which have antioxidant, antiaging, and anti-inflammatory functions; the anthocyanin biosynthesis pathway has been widely investigated in many plants [90,91,92]. Previous studies have demonstrated that anthocyanin regulated the plant petal color formation and influenced pigmentation [93,94]. It has been shown that six anthocyanins (cyanidin, delphinidin, malvidin, pelargonidin, peonidin, and petunidin) are common in petal color changes [95,96]. In our research, 18 differential anthocyanins were identified in the *L. radiata* petals and mainly contained three categories of cyanidin, peonidin, and pelargonidin (Figure 7A). Additionally, most anthocyanin compounds showed significantly high expressions in the red petals; pelargonidin (pelargonidin-3-O-glucoside-5-O-arabinoside) may especially be the main source of red petal formation. Similar results have also been acknowledged in previous academic studies [95,96,97]. Furthermore, some key structural genes (such as *DFR*, *4CL*, *CHI*, *ANS*, *CHS*, and *F3′H*) involved in anthocyanin biosynthesis were also found based on RNA-seq profiling and were significantly up-regulated, indicating that WRKY TFs have a greater effect on anthocyanin [53].

In recent years, many studies have investigated the conservation of WRKY-based regulatory mechanisms in the anthocyanin biosynthesis pathway, such as in *Arabidopsis*, in which WRKY TFs combined with the MBW (MYB-bHLH-WD40) complex to regulate the anthocyanin biosynthesis pathway and control flavonoid pigment pathways [98,99]. In addition, WRKY was found to be involved in rhododendron color formation [100]. These studies illustrate that WRKY TFs play an important role in the pigment pathway. In our research, *LrWRKY3* showed that significantly different expression levels in petals might be candidate regulators of anthocyanin accumulation in petal color changes in *L. radiata* (Figure 7 and Figure 8). The gene—metabolite correlation network showed that the metabolite pelargonidin-3-O-glucoside-5-O-arabinoside was strongly correlated with *LrWRKY3* in the anthocyanin biosynthesis pathways (Figure 7). Interestingly, our results also showed that the LrWRKY3 transcription factor specifically interacted with the *ANR* and *LAR* gene, suggesting that it is involved in petal coloration in *L. radiata* (Figure 8). The WRKY transcription factors participate in the synthesis of anthocyanins across a diversity of different species; therefore, more research is needed to understand how WRKY has evolved throughout angiosperms and what function it serves in other lineages. 

## 4. Materials and Methods

### 4.1. Plant Materials and Plant Treatments

*Lycoris radiata* (L′Her.) Herb. was planted in the Experimental Plantation of the Institute of Botany, Jiangsu Province and the Chinese Academy of Sciences, Nanjing, China. The seedlings with the same or similar sizes (2.8–3.2 cm) in diameter were transferred into plastic pots with a mixture of vermiculite and soil (1:1, *v*/*v*) and maintained in a plant growth chamber under the following conditions: 16 h light/8 h dark cycle at 25 °C/22 °C and 120 μmol m^−2^ s^−1^ irradiation). After one week of maintenance, the seedlings were subjected to 100 μmol L^−1^ methyl jasmonate (MeJA) for 0 h, 6 h, 12 h, 24 h, and 36 h and drought stress (20% PEG6000) for 24 h. Each treatment was replicated three times. The tissue-specific transcription profiles of the *LrWRKY* genes were explored in the petals, flower-stalks, gynoeciums, stamens, leaves, seeds, roots, and bulbs of these plants. All the samples were immediately frozen in liquid nitrogen and stored at −80 °C.

### 4.2. Transcriptome-Wide Identification and Expression Profiling of LrWRKY Genes

The *L. radiata* transcriptome database during the four flower development stages with 87,584 unigenes was used for potential LrWRKY searching [53]. The AtWRKY proteins downloaded from TAIR (Arabidopsis Information Resource database, https://www.arabidopsis.org/ (accessed on 20 September 2022)) were utilized to determine the sequence homology with *L. radiata* transcripts from the database using basic local alignment (BLASTn). The hidden Markov model (HMM) profile of the WRKY DNA binding domain (PF03106) obtained from the Pfam protein family database (http://pfam.xfam.org (accessed on 20 September 2022)) [101] was used to search candidate *LrWRKY* genes. Then, we verified the WRKY domain in the predicted *LrWRKY* transcription factors utilizing the NCBI Batch CD-Search Tool (https://www.ncbi.nlm.nih.gov/Structure/bwrpsb/bwrpsb.cgi (accessed on 20 September 2022)) with default parameters. This characteristic was deemed to have a high-confidence association with the conserved domain. The sequences predicted as specific hits were retained for further analysis (Appendix A). Furthermore, the PFAM and SMART (http://smart.embl-heidelberg.de/ (accessed on 20 September 2022)); [102]) databases were used for verifying the WRKY domain in all the candidate protein sequences. Finally, the ExPASy website (https://web.expasy.org/protparam (accessed on 20 September 2022)) [103]) was utilized to determine the full length of the amino acid sequences, isoelectric points (PI), molecular sizes (MW), and protein instability index.

### 4.3. Phylogenetic Tree and Protein Motif Analyses of LrWRKY Proteins

The phylogenetic tree of WRKYs from *L. radiata* and *A. thaliana* was constructed with MEGA7 using the neighbor-joining method (https://www.megasoftware.net/ (accessed on 20 September 2022)). The LrWRKY proteins were then classified according to their phylogenetic relationships with AtWRKY proteins. The online tool MEME (Multiple EM for Motif Elicitation, version 5.1.1) was utilized to search for the conserved motifs of LrWRKY proteins (https://meme-suite.org/meme/tools/meme; [104] (accessed on 20 September 2022)), with the motif number and a width of 21–50 for each gene. The motifs were also searched in protein databases using the SMART program (http://smart.embl.de/ (accessed on 20 September 2022)).

### 4.4. Gene Ontology Annotation of LrWRKYs

The Gene Ontology (GO) functional annotations were conducted using KOBAS. The top 20 functional terms for credibility in the three categories (biological processes, cellular components, and molecular function) were selected for visualization.

### 4.5. Expression Profiles Analysis and Real-Time Quantitative PCR Analysis of LrWRKYs Genes

The RNA-seq data for the *LrWRKY* genes were obtained from previous studies on gene expression in different flower developmental stages and MeJA treatment [53,54]. The *LrWRKY* gene expression profiles were evaluated utilizing the values of FPKM. TBtools [105] software was utilized to generate *LrWRKYs* expression heatmaps. The total RNA was extracted using an RNA prep Pure Plant Kit (Cat BC508, Huayueyang, Beijing, China) according to the manufacturer’s protocol. The cDNA was synthesized by using a PrimeScript™ II 1st Strand cDNA Synthesis Kit (Takara Bio, Dalian, China) and utilized for qRT-PCR assays. The relative expression levels of the genes were analyzed using qRT-PCR with SYBR^®^ Premix Ex Taq™ II (Takara Bio, Dalian, China) on a Bio-Rad iQ5 Real-Time PCR System (Bio-Rad, Hercules, CA, USA) in 15 μL reactions. Each reaction contained 7.5 μL 2 × TransStart^®^ Top Green qPCR SuperMix, 5.9 μL ddH2O, 1μL cDNA, and 0.6 μL of 10 μM forward and reverse primers, and 6.2 μL of ddH2O. The RT-qPCR protocol included the following: the PCR reaction conditions were at 95 °C for 5 min; denaturation 5 s at 95 °C; 60 °C for 30 s; and 40 cycles. The *LrTIP41* gene was utilized as an endogenous control to normalize the relative expression levels based on the 2^−ΔΔCt^ method [106]. The specific primer sequences used in this research are listed in Appendix A.

### 4.6. GenesGene Cloning and Construction of Expression Vectors

*The LrWRKYs* were cloned based on putative ORFs of unigenes from the RNA-seq database. The primers were synthesized for ORF sequence amplification using Tks GflexTM DNA Polymerase (Takara, Dalian, China) from *L. radiata* petal cDNA (Appendix A). The reaction conditions were: 3 min of 95 °C, 40 cycles for 30 s at 94 °C, 30 s at 58 °C, 2 min at 72 °C, and with extension at 72 °C for 10 min. The PCR products were cloned into pTOPO001 simple vectors (Genesand, Beijing, China). Afterward, those T-vectors were transferred into TOP10 competent cells (Genesand, Beijing, China) for amplification. The *LrWRKYs* overexpression vectors were built by linking their ORFs into a pBIN-MCS-GFP4 plant transformation vector by using the One Step Cloning Kit (Genesand, Beijing, China). Then, the *35S:LrWRKYs* recombinant vectors were transformed into *Agrobacterium tumefaciens* GV3101 competent cells.

### 4.7. Anthocyanin Quantification and Data Analysis of L. radiata

The metabolite data acquisition was conducted with a UPLC-ESI-MS/MS system (Shim-pack UFLC SHIMADZU Nexera X2) (ultrafast liquid chromatography) instrument coupled to an MS/MS (Applied Biosystems 4500 QTRAP). The chromatographic separation was carried out on a Waters ACQUITY UPLC HSS T3 system (Shim-pack UFLC SHIMADZU CBM30A, see text footnote 2), which was equipped with a C18 column (1.8 mm × 2.1 mm × 100 mm), at 40 °C. The injection volume was 5 mL at a flow rate of 0.4 mL/min. The organic phase was acetonitrile (0.1% formic acid) and the mobile phase was ultrapure water (0.1% formic acid,). The elution gradient was as follows: 0 min, 95:5 water/acetonitrile (*v*/*v*); 9.0 min, 5:95 water/acetonitrile; 10.0 min, 5:95 water/acetonitrile; 11.1 min, 95:5 water/acetonitrile; and 14.0 min, 95:5 water acetonitrile. The metabolites eluted from the HPLC were monitored for each period using scheduled multiple reaction monitoring (MRM). The MRM signals were transformed and analyzed using the Analyst 1.6.3 software (Metware Biotechnology Co., Ltd., Wuhan, China). The metabolite identification and quantification were conducted following the commercially available standard Metabolites Database [107] (Metware Biotechnology Co., Ltd., Wuhan, China) and public metabolite databases [108]. The transcriptional regulatory networks were generated by combining the Pearson correlation coefficient (|PCC| > 0.8 or |PCC| > 0.9) between structural genes and transcription factors and anthocyanins metabolites. Finally, the regulatory network maps were visualized by Cytoscape (v.3.7.2, Seattle, WA, USA).

### 4.8. Subcellular Localization Analysis of LrWRKY Proteins

The subcellular localization of LrWRKY proteins was predicted using WolfPsort (https://wolfpsort.hgc.jp (accessed on 20 September 2022); [109]), ProtComp 9.0 (http://linux1.softberry.com (accessed on 20 September 2022)), and Plant-mPLoc (http://www.csbio.sjtu.edu.cn/bioinf/plant-multi/ (accessed on 20 September 2022)). The coding regions of each *LrWRKY* gene were inserted into the pBinGFP4 plant expression vector. This vector was then used to transform *A. tumefaciens* strain GV3101 bacteria. The recombinant *A. tumefaciens* strains that contained various constructs were cultivated, harvested, followed by resuspension in an invasive solution (10 mM MES, 0.2 mM Acetosyringone, and 10 mM MgCl_2_) with a final OD600 value of 0.6. Forty-day-old *N. benthamiana* plants were used for infiltration. After infiltration, the plants were grown at 22 °C in the dark and then transferred to normal growth conditions (25 °C/16 h light and 22 °C/8 h dark cycle) for three days. The GFP fluorescent signals in the epidermal cells of *N. benthamiana* leaves were observed under a confocal microscope (Zeiss LSM900, Jena, Germany).

### 4.9. PPI Network Prediction of LrWRKY Proteins

The potential protein–protein interaction networks were predicted using the STRING online database (version 11.5) based on *A. thaliana* homologous proteins (https://cn.string-db.org (accessed on 20 September 2022); [110]). The protein sequences of 31 LrWRKYs (Appendix A) were uploaded into the server selecting *A. thaliana* as the comparative organism. The *LrWRKY* gene interaction network was constructed after blasting with the highest bitscore.

### 4.10. Statistical Analysis

All the experiments were independently duplicated at least three times. The results were represented as mean ± SD of biological triplicates. The student *t*-test was utilized for data analysis. * *p* < 0.05 and ** *p* < 0.01 were regarded significant.

## 5. Conclusions

In this study, 31 *LrWRKY* genes were identified from the transcriptome of *L. radiata* and enriched by systematically analyzing the basic biochemical information, phylogenetic relationships, conserved motifs, and gene expression profiling. Most *LrWRKY* genes were revealed to likely play various important roles in *Lycoris* growth and development, especially in flower development stages. The analysis of the expression pattern of *LrWRKY* genes in response to drought stress and MeJA treatment suggested that the *LrWRKY* genes could be importantly involved in drought stress as well as in JA signaling pathway. The gene—metabolite correlation network analysis revealed a potential regulatory role of *LrWRKYs* in anthocyanin biosynthetic pathway. A PPI analysis of LrWRKY proteins was performed to explore the functions and regulatory mechanisms in *L. radiata*. In conclusion, this study provides valuable information for further research on the regulatory mechanisms of WRKY TFs in plant growth, secondary metabolism, and resistance to various stressors.

## Figures and Tables

**Figure 1 ijms-24-02423-f001:**
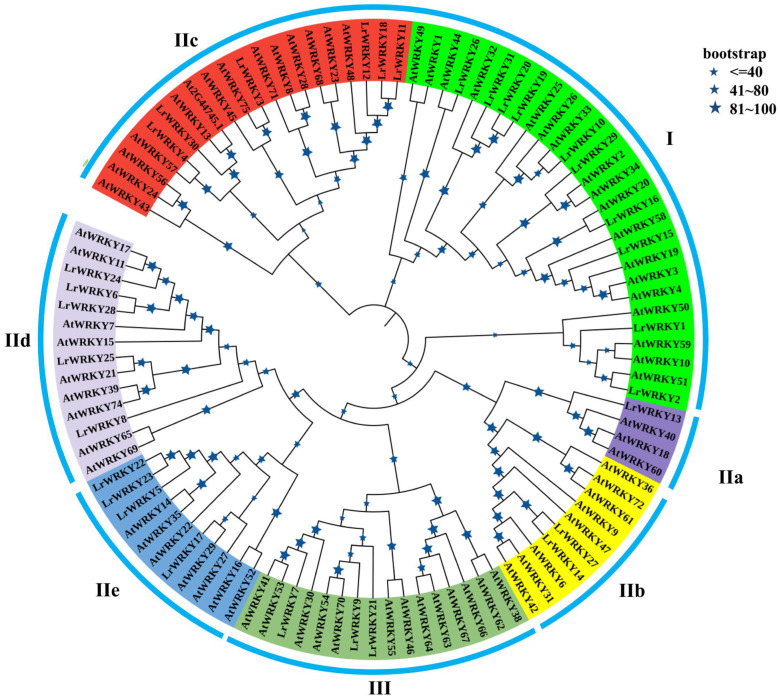
The phylogenetic analysis of *L. radiata* WRKY proteins. The neighbor-joining method was utilized to generate the phylogenetic tree relying on the WRKY domains alignment. These numbers are computed using 1000 bootstrap replicates to verify reliability. The replicate tree percentages indicated with different sizes of asterisk are presented on branches. The tree shows the three subfamilies marked with blue font on a colored background.

**Figure 2 ijms-24-02423-f002:**
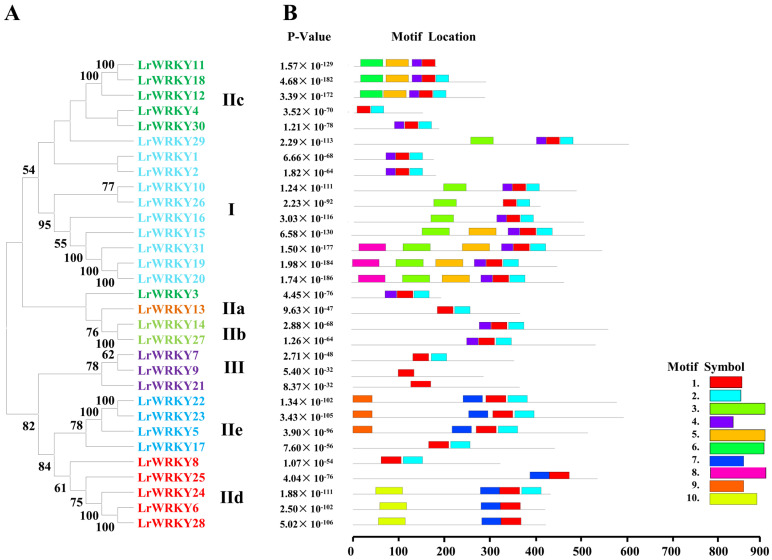
The phylogenetic relationships and conserved motifs analysis of LrWRKY proteins. (**A**) Neighbor-joining LrWRKYs phylogenetic tree (bootstrap values for 1000 replicates); (**B**) Conserved motif distribution in LrWRKY proteins. Different motifs are represented with different colored boxes. The motif length is represented by the box length.

**Figure 3 ijms-24-02423-f003:**
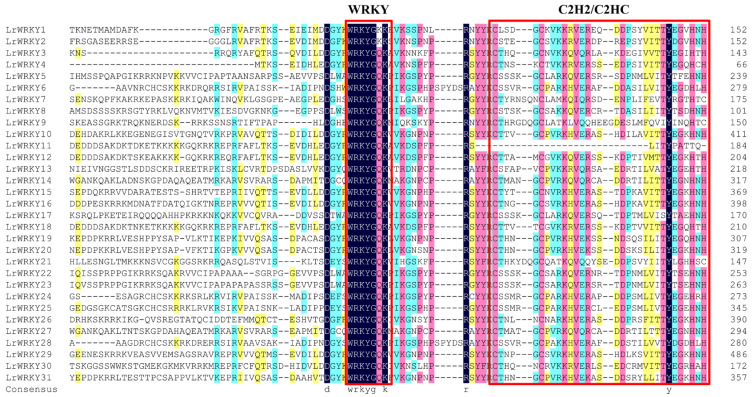
Sequence alignment of 31 LrWRKY proteins. A screenshot of the MEGA X version 10.2.6, showing variation in the conserved domain (WRKYGQK) of LrWRKY proteins and the variation of the conserved domain is encircled in different colors. The yellow boxes indicate 30% identity of amino acids, the blue boxes indicate 50% identity of amino acids, the red boxes indicate 75% identity of amino acids, and the black boxes indicate 100% identity of amino acids.

**Figure 4 ijms-24-02423-f004:**
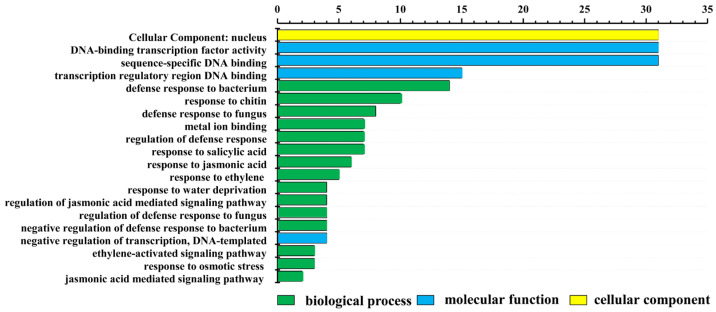
Gene ontology (GO) enrichment analysis of LrWRKY proteins. The top 20 GO terms of level 2 in biological process, cellular component, and molecular function were visualized. The X and Y axes represent the enriched protein numbers and the information on GO terms, respectively.

**Figure 5 ijms-24-02423-f005:**
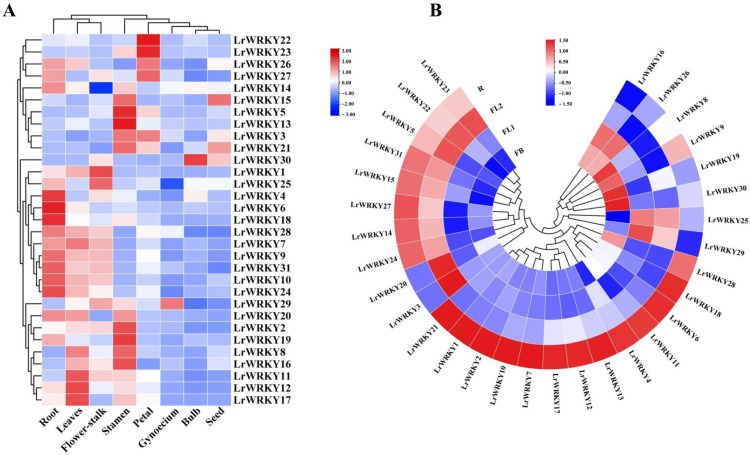
*LrWRKY* gene expression patterns in various *L. radiata* tissues. (**A**) Expression profile heatmap with hierarchal clustering of *LrWRKYs* in different tissues of *L. radiata*. (**B**) Expression profile heatmap with hierarchal clustering of *LrWRKYs* at different flower developmental stages of *L. radiata*. The relative expression for each gene is depicted by color intensity in each field. Higher values are represented by red, whereas lower values are represented by blue. FB, flower—bud differentiation stage; FL1, partially opening flower stage; FL2, fully opened flower stage; R, senescent flower stage.

**Figure 6 ijms-24-02423-f006:**
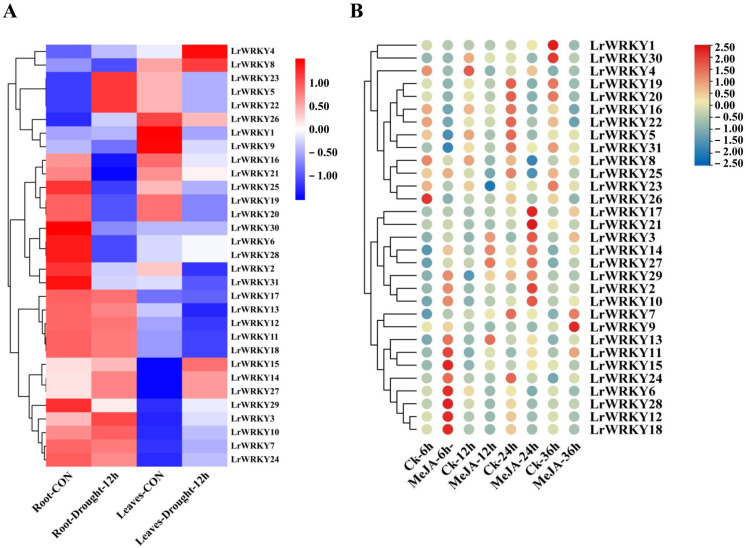
*LrWRKY* gene expression profiles under drought stress and MeJA treatment in *L. radiata.* (**A**) Heatmap with hierarchical cluster analysis of drought-responsive differentially expressed *LrWRKY* genes. (**B**) Heatmap of *LrWRKY* genes expression profiles with MeJA treatment. Red and blue represent high and low relative transcript abundance, respectively.

**Figure 7 ijms-24-02423-f007:**
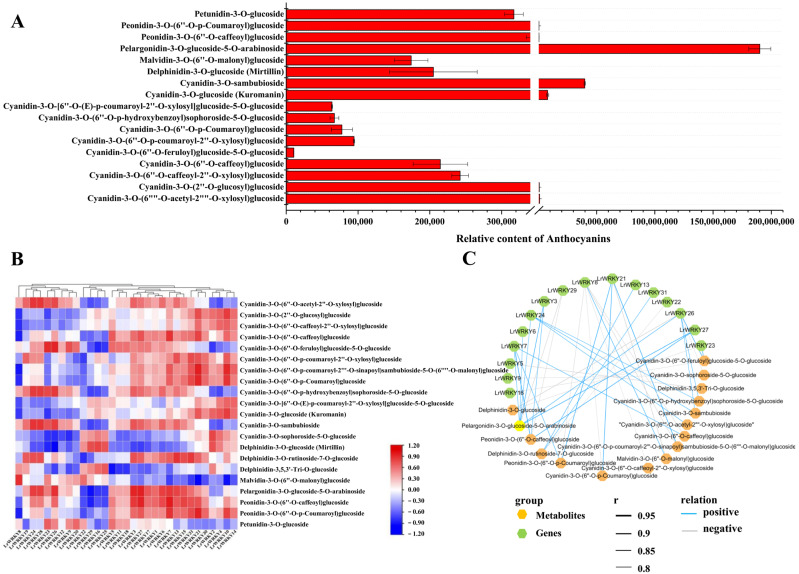
The regulatory network of key anthocyanins metabolites and *LrWRKY* genes in *L. radiata.* (**A**) Eighteen anthocyanins were identified and quantified from petals of *L. radiata.* (**B**) The Pearson’s correlation coefficients of *LrWRKYs* with an anthocyanin biosynthesis pathway in *L. radiata*. Higher values are represented by red, whereas lower values are represented by blue. (**C**) Correlation network of metabolite-related *LrWRKY* genes involved in anthocyanin biosynthetic pathways. r represents the Pearson correlation coefficient; relation represents the correlation, including positive (r > 0.8) and negative correlations (r < −0.8).

**Figure 8 ijms-24-02423-f008:**
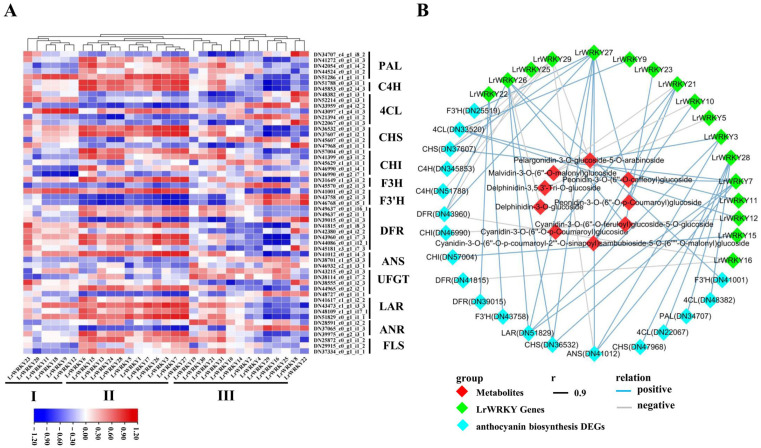
The regulatory network of key differently expressed genes (DEGs) of anthocyanin biosynthesis, metabolites, and *LrWRKY* genes in *L. radiata*. (**A**) The Pearson’s correlation coefficients of *LrWRKYs* with DEGs of anthocyanin biosynthesis in *L. radiata*. The enzymes include cinnamate 4-hydroxylase (C4H), 4-coumarate:CoA ligase (4CL), phenylalanine ammonia lyase (PAL), chalcone synthase (CHS), flavone 3-hydroxylase (F3H), chalcone isomerase (CHI), flavonoid 3′-hydroxylase (F3′H), dihydroflavonol reductase (DFR), flavonol synthase (FLS), UDP-flavonoid glucosyl transferase (UFGT), anthocyanidin synthetase (ANS), anthocyanidin reductase (ANR), and leucoanthocyanidin reductase (LAR). (**B**) Correlation network of metabolite-related *LrWRKY* genes and DEGs involved in anthocyanin biosynthetic pathways. r represents the Pearson correlation coefficient; relation represents the correlation, including positive (r > 0.9) and negative correlations (r < −0.9).

**Figure 9 ijms-24-02423-f009:**
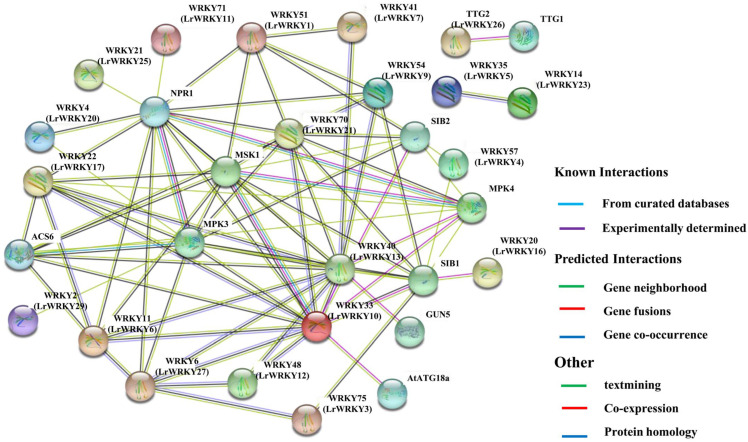
The predicted network of protein–protein interactions between LrWRKYs using the STRING database. Different colors represent different interaction types. *Arabidopsis* WRKY names are marked, whereas their homologs in *L. radiata* are in parentheses.

**Figure 10 ijms-24-02423-f010:**
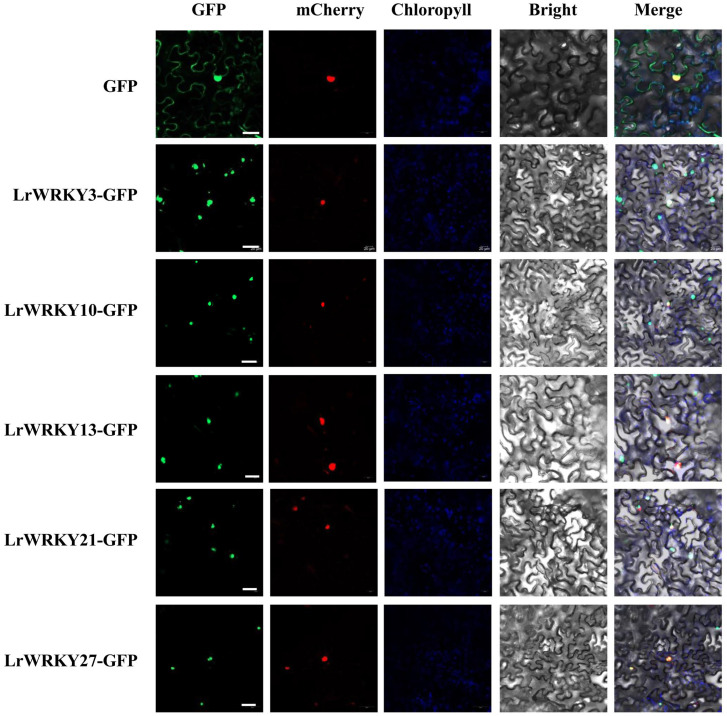
Subcellular localization of LrWRKYs with GFP as a control, which were transiently expressed in *N. benthamiana* leaves. The photographs were taken using the green channel (GFP fluorescence), red channel (nuclear marker protein AtCOL), blue channel (chlorophyll represents chlorophyll auto fluorescence), bright channel, and their combination under a confocal microscope. Merged bright-field, green fluorescence, red fluorescence, blue fluorescence, and green—red—blue fluorescence images. Scale bar = 10 μm.

## Data Availability

All data are displayed in the manuscript.

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
