# Peer review of "Characterization of the *WRKY* Gene Family Related to Anthocyanin Biosynthesis and the Regulation Mechanism under Drought Stress and Methyl Jasmonate Treatment in *Lycoris radiata"

_ijms, 2023, doi:10.3390/ijms24032423_

Round 1
Reviewer 1 Report
The manuscript entitled "Characterization of the WRKY gene family related to anthocyanin biosynthesis and the regulation mechanism under drought stress and methyl jasmonate treatment in Lycoris radiata" presents a nice mosaic of experiments that adds to current knowledge of anthocyanin biosynthesis. The work is well designed, the results are well presented and the overall message to the broad scientific community is clear and sound.
Author Response
MS ID: ijms-2115434
MS title: Characterization of the WRKY gene family related to anthocyanin biosynthesis and the regulation mechanism under drought stress and methyl jasmonate treatment in Lycoris radiata
Dear Jankin Jia, Editor of International Journal of Molecular Sciences:
We appreciate you and the reviewers to review our manuscript and gave us valuable suggestions on how to improve it. We carefully address all of your concerns and comments, and additional revision has been performed. Please see the detailed revision in our revised manuscript.
In summary, we appreciate greatly the comments proposed by you and the respected reviewers, and we have revised the manuscript according to your useful suggestions. The revised manuscript was re-uploaded, and all revised sections were highlighted. We also wish that our answer could be as complete as possible to the questions of reviewers and hope that the revised manuscript might be acceptable for possible publication in International Journal of Molecular Sciences.
We thanks for your consideration.
Sincerely yours,
Dr. Zhong Wang
Email: wangzhong@cnbg.net
Fax & Tel: +86-25-84347055
Institute of Botany
Jiangsu Province and Chinese Academy of Sciences
Nanjing 210014
Jiangsu Province
China
Revision Notes
Independent Review Report, Reviewer 1
Reviewer 1
Question 1: The manuscript entitled "Characterization of the WRKY gene family related to anthocyanin biosynthesis and the regulation mechanism under drought stress and methyl jasmonate treatment in Lycoris radiata" presents a nice mosaic of experiments that adds to current knowledge of anthocyanin biosynthesis. The work is well designed, the results are well presented and the overall message to the broad scientific community is clear and sound.
Answer: Thank you very much for your kind comments. Other corrections we found during re-check have been made as well.
Dr. Zhong Wang
Email: wangzhong@cnbg.net
Fax & Tel: +86-25-84347055
Institute of Botany
Jiangsu Province and Chinese Academy of Sciences
Nanjing 210014

Author Response
MS ID: ijms-2115434
MS title: Characterization of the WRKY gene family related to anthocyanin biosynthesis and the regulation mechanism under drought stress and methyl jasmonate treatment in Lycoris radiata
Dear Jankin Jia, Editor of International Journal of Molecular Sciences:
We appreciate you and the reviewers to review our manuscript and gave us valuable suggestions on how to improve it. We carefully address all of your concerns and comments, and additional revision has been performed. Please see the detailed revision in our revised manuscript.
In summary, we appreciate greatly the comments proposed by you and the respected reviewers, and we have revised the manuscript according to your useful suggestions. The revised manuscript was re-uploaded, and all revised sections were highlighted. We also wish that our answer could be as complete as possible to the questions of reviewers and hope that the revised manuscript might be acceptable for possible publication in International Journal of Molecular Sciences.
We thanks for your consideration.
Sincerely yours,
Dr. Zhong Wang
Email: wangzhong@cnbg.net
Fax & Tel: +86-25-84347055
Institute of Botany
Jiangsu Province and Chinese Academy of Sciences
Nanjing 210014
Jiangsu Province
China
Revision Notes
Independent Review Report, Reviewer 2
In page 239, the authors claim that “the expression patterns of LrWRKY7, LrWRKY9, LrWRKY10, LrWRKY24, LrWRKY28, and 239 LrWRKY31 were similar suggested that these genes may have similar functions”.
QUERY: does it mean they have similar functions or perhaps they are under the same regulatory constraints? Have you looked at the promoter region to see if they are more similar or share similar CREs?Could it be said that they respond to the same signals instead?
Answer: Thank you very much for the critical comments. According to the suggestion, we have revised these sentences, which were also listed as below:
“In particular, we noticed that similar expression patterns for LrWRKY7, LrWRKY9, LrWRKY10, LrWRKY24, LrWRKY28, and LrWRKY31 in different tissues suggest possible redundancy”. In addition, because of lacking the genome information resources of Lycoris species, the cis-acting element prediction in the promoters of the LrWRKY genes was not done. Nonetheless, we have attempted to obtain the promoter sequence of LrWRKY genes with self-formed adaptor PCR (Wang et al., 2007). However, we failed. Thus, we would like to continue PCR-amplifying the promoter sequence of LrWRKY genes with new method to check whether they share similar cis-acting elements in the future work.
Reference
Wang, S., He, J., Zhang, F., Cui, Z., and Li, S. (2007). Self-formed adaptor PCR: a simple and efficient method for chromesome. Appl. Environ Micro. 2007; 73. https://doi.org/10.1128/AEM.02973-06
Line 470 – from instead of form
Answer: Thank you very much for your kind suggestion. We have revised the term “form” in Line 470 to “from”. Please check it in the revised manuscript.
Line 541 – Anthocyanins is the key metabolites…switch to Anthocyanins is among the key metabolites
Answer: Thank you very much for your kind suggestion. We have revised the sentence. Please check it in the revised manuscript.
Line 549 – you meant to say “almost all”? it is confusing as it is…Comments:
Answer: Thank you very much for your kind suggestion. We have revised the term “almost all”, please check it in the revised manuscript.
Finally, we want to thank you sincerely for your useful suggestions. Other corrections we found during re-check have been made as well.
Dr. Zhong Wang
Email: wangzhong@cnbg.net
Fax & Tel: +86-25-84347055
Institute of Botany
Jiangsu Province and Chinese Academy of Sciences
Nanjing 210014
